# Single Dose of Recombinant Chimeric Horsepox Virus (TNX-801) Vaccination Protects Macaques from Lethal Monkeypox Challenge

**DOI:** 10.3390/v15020356

**Published:** 2023-01-26

**Authors:** Ryan S. Noyce, Landon W. Westfall, Siobhan Fogarty, Karen Gilbert, Onesmo Mpanju, Helen Stillwell, José Esparza, Bruce Daugherty, Fusataka Koide, David H. Evans, Seth Lederman

**Affiliations:** 1Department of Medical Microbiology & Immunology, Li Ka Shing Institute of Virology, University of Alberta, Edmonton, AB T6G 2R3, Canada; 2Southern Research, Birmingham, AL 35205, USA; 3Tonix Pharmaceuticals, Dartmouth, MA 02748, USA; 4LINQ Pharma Consulting Inc., Aldie, VA 20105, USA; 5Institute of Human Virology, University of Maryland School of Medicine, Baltimore, MD 21201, USA

**Keywords:** horsepox, monkeypox, smallpox, immune responses, vaccinia

## Abstract

The ongoing global Monkeypox outbreak that started in the spring of 2022 has reinforced the importance of protecting the population using live virus vaccines based on the vaccinia virus (VACV). Smallpox also remains a biothreat and although some U.S. military personnel are immunized with VACV, safety concerns limit its use in other vulnerable groups. Consequently, there is a need for an effective and safer, single dose, live replicating vaccine against both viruses. One potential approach is to use the horsepox virus (HPXV) as a vaccine. Contemporary VACV shares a common ancestor with HPXV, which from the time of Edward Jenner and through the 19th century, was extensively used to vaccinate against smallpox. However, it is unknown if early HPXV-based vaccines exhibited different safety and efficacy profiles compared to modern VACV. A deeper understanding of HPXV as a vaccine platform may allow the construction of safer and more effective vaccines against the poxvirus family. In a proof-of-concept study, we vaccinated cynomolgus macaques with TNX-801, a recombinant chimeric horsepox virus (rcHPXV), and showed that the vaccine elicited protective immune responses against a lethal challenge with monkeypox virus (MPXV), strain Zaire. The vaccine was well tolerated and protected animals from the development of lesions and severe disease. These encouraging data support the further development of TNX-801.

## 1. Introduction

The subfamily *Chordopoxvirinae* in the family *Poxviridae* is comprised of enveloped viruses with large genomes comprising a linear double-stranded DNA molecule of 128 to 270 kilobase pairs (kbp). Two members of the family, variola (VARV) and monkeypox virus (MPXV), are important human pathogens that can cause fatal human disease. VARV is the causative agent of human smallpox with mortality rates of 10–30%. Although smallpox was declared eradicated in 1980 following a global immunization campaign [1], monkeypox, a zoonotic disease, is emerging as a significant threat globally [2]. Monkeypox Zaire or Clade I, causes a human disease similar to smallpox with a mortality rate of 10%.

Currently, there are two Food and Drug Administration (FDA) approved smallpox vaccines, one of which is also indicated for monkeypox. The ACAM2000^®^ live replicating vaccinia vaccine was approved in 2007 and is indicated for active immunization against smallpox in persons determined to be at high risk for VARV infection [3]. In September 2019, the FDA approved Modified Vaccinia Ankara by Bavarian Nordic (MVA-BN (Jynneos^TM^), a replication-deficient vaccinia vaccine, indicated for the prevention of smallpox and monkeypox in adults 18 years of age and older determined to be at high risk for smallpox or MPXV infection [4].

However, there are certain potential limitations of ACAM2000 and MVA-BN. The FDA-approved label for ACAM2000 carries a boxed warning of potentially serious adverse events such as myocarditis and/or pericarditis. The rate of cardiac adverse events (AEs) was estimated at 5.7 per 1000 [5]. A 2015 prospective study of the incidence of myocarditis/pericarditis and new onset cardiac symptoms following smallpox vaccination, concluded that there was a four-fold higher incidence of new-onset cardiac symptoms post-smallpox vaccination compared to post-influenza immunization [6]. The molecular mechanism of ACAM2000-induced cardiotoxicity remains unclear. MVA-BN has not been studied in enough patients to determine if the risk of cardiotoxicity is lower than ACAM2000. ACAM2000 is contraindicated in immunodeficient individuals because of an increased risk of disseminated vaccinia.

The FDA approved MVA-BN in 2019 based on the demonstration of immunologic non-inferiority to ACAM2000 in humans (peak serum neutralizing antibody titers) and protection against the MPXV challenge in a non-human primate (NHP) model [7,8]. The live replicating vaccinia virus (VACV) vaccines such as ACAM2000 are administered by a single percutaneous procedure with a bifurcated needle, whereas MVA-BN is administered by subcutaneous injection and requires two injections separated by a month or more to induce acceptable neutralizing of antibody titers. A single pivotal effectiveness study was conducted [9]. In this study, a comparison of geometric mean neutralizing antibody titers (GMTs) between MVA-BN and ACAM2000 at “peak visits” (defined as 2 weeks after the second MVA-BN vaccination and 4 weeks after the ACAM2000 vaccination) demonstrated non-inferiority in the per-protocol study population. Although, MVA-BN is a safer vaccine, the vaccine regimen one month apart is a potential serious draw back in an outbreak setting.

Since the eradication of smallpox in 1980, the risk and reward ratio of vaccination against smallpox has changed. At present, the risks associated with universal ACAM2000 vaccination of the general population may outweigh the potential benefits. In most of the United States (U.S.), routine vaccination against smallpox ended in the early 1970s. ACAM2000 is currently used primarily in U.S. military personnel deployed to areas that the U.S. government has determined to be at increased risk of deliberate smallpox exposure and in laboratory workers at potential occupational risk for exposure to orthopoxviruses. Safety concerns associated with the vaccine, limit its further use in groups such as first responders.

The threat of the intentional or accidental release of VARV and the reemergence of smallpox has not completely disappeared. In addition, the 2022 outbreak of monkeypox highlights the continuing need to develop safer single dose smallpox and monkeypox vaccines [10]. One potential approach to address this unmet need is to explore the use of the horsepox virus (HPXV) as a vaccine. HPXV was extensively used in the 19th century for smallpox vaccination [11,12,13] and it has been proposed that contemporary VACV share a common ancestor with HPXV [14,15,16,17,18].

Since HPXV may be extinct and the only existing isolate is not available for investigation, we recently developed a recombinant chimeric HPXV (rcHPXV, known by the commercial name of TNX-801) as a vaccine platform and for the prevention of smallpox and MPXV disease [19]. The TNX-801 vaccine was constructed using large-scale gene synthesis and poxvirus reactivation strategies. The design of the TNX-801 genome was based on the published sequence for HPXV strain MNR-76 (GenBank accession DQ792504) [20]. TNX-801 may possess advantages for use in healthy, immunocompetent non-pregnant individuals, without a history of eczema or cardiac disease, and as part of a public health vaccination policy to respond to an event of VARV reintroduction.

Modern vaccinia vaccines have characteristic deletions near the boundaries of the left and right inverted terminal repeat (ITR) sequences of their genomes relative to other related orthopoxviruses such as HPXV (Figure 1) [15,21]. Consequently, HPXV has additional genes (relative to VACV vaccines), encoded by “complete” left and right ITRs, that may attenuate immune reactogenicity, account for small plaque size in vitro [22], and may also serve as antigens for protective immune responses.

In this proof-of-concept report, we investigated the potential of TNX-801 as a vaccine against a lethal form of monkeypox by investigating the immunogenicity and efficacy in a cynomolgus macaque model.

## 2. Materials and Methods

### 2.1. Viruses and Cells

Shope fibroma virus, African green monkey cells (BSC-40), and Buffalo green monkey kidney cells (BGMK) were originally obtained from American Type Culture Collection. Complete genome sequencing of a laboratory isolate of VACV was obtained following plaque purification of a VACV isolate similar to VACV strain ACAM2000. Illumina next generation sequencing of this isolate included the terminal hairpin sequences as well as additional repeat regions within the inverted terminal repeat and has been deposited in the NCBI Genbank database (Accession # MN974380). The viruses and cells were cultured in minimum essential media (MEM) supplemented with L-glutamine, nonessential amino acids, sodium pyruvate, and 5% fetal bovine serum (FBS) (Thermo Fisher Scientific, Mississauga, ON, Canada). All cell lines were regularly screened for mycoplasma using the VenorGem PCR detection kit (Sigma Millipore, Oakville, ON, Canada).

Recombinant chimeric HPXV (rcHPXV) and vaccinia (rVACV) were utilized as vaccines. The methods for the generation of rcHPXV and rVACV have been previously described [19,23]. The rVACV genome sequence was deposited into GenBank (Accession # MN974381).

### 2.2. Virus Stocks

All rcHPXV and rVACV stocks were grown and titrated on BSC-40 cells. All NHP experiments used viruses prepared by releasing viruses from the infected cells using a Dounce homogenizer in 10 mM Tris-HCl pH 9 and 2 mM MgCl_2_ buffer. The virus lysate was treated with 50 U/mL benzonase (Sigma Millipore, Oakville, ON, Canada) at 37 °C for 30 min. The virus preparations were subsequently purified and concentrated by centrifugation at 26,500× *g* for 90 min onto a 36% sucrose cushion. Viruses were resuspended in 10 mM Tris pH 8 and stored at −80 °C. Virus stocks were initially titrated on BSC-40 cell lines to determine virus concentrations at the University of Alberta. These titers were used to calculate the original dose of each virus to be used to inoculate cynomolgus macaques. Following inoculation of the cynomolgus macaques, the viruses were re-titrated on Vero cells to obtain a dose estimate of virus that went into each animal.

### 2.3. Ethics Statement

This work was supported by an approved Institute Animal Care and Use Committee (IACUC) animal research protocol. The research was conducted under an IACUC approved protocol in compliance with the Animal Welfare Act, PHS Policy, and other Federal statutes and regulations relating to animals and experiments involving animals. The facility where this research was conducted is accredited by the Association for Assessment and Accreditation of Laboratory Animal Care (AAALAC International) and adheres to principles stated in the Guide for the Care and Use of Laboratory Animals, National Research Council, 2011 [24].

### 2.4. Nonhuman Primate Study Design

A total of 16 (8 males, 8 females) cynomolgus macaques (*Macaca fascicularis*) of Chinese origin aged 5–8 years and weighing ~3–9 kg were obtained. All NHPs were prescreened and determined to be negative for herpes B virus, simian T-lymphotropic virus 1, simian immunodeficiency virus, simian retrovirus D 1/2/3, tuberculosis, *Salmonella* spp., *Campylobacter* spp., hypermucoviscous *Klebsiella* spp., Shigella spp, MPXV, and VACV.

### 2.5. Immunization Procedure

NHPs were vaccinated on day 0 via scarification inoculation with identified doses for each group (Figure 2). Using virus titers obtained at the University of Alberta on BSC-40 cells, four NHPs per group were vaccinated at Day 0 with 6.2 log_10_ plaque forming units (PFU) of TNX-801 (high dose), 5.4 log_10_ PFU of TNX-801 (low dose), 5.4 log_10_ PFU of rVACV, or mock vaccinated (Figure 2). After inoculation, the viral stocks were retirated on Vero cells and the administered doses were calculated to be: 6.6 log_10_ PFU of TNX-801 (high dose), 5.7 log_10_ PFU of TNX-801 (low dose), 5.0 log_10_ PFU of rVACV, or mock vaccinated (Figure 2). Briefly, the vaccination site between the shoulder blades was shaved and cleaned. To simulate percutaneous administration in humans, about 10 μL of vaccine or vehicle was placed on the skin using a sterile pipette tip, and a bifurcated needle was used to scarify the area by penetrating the skin vertically approximately 15 times. The residual vaccine was removed using a sterile gauze. One animal in the TNX-801 (low dose) and three animals in the rVACV did not produce a major cutaneous reaction at the vaccination site (a “take”) by day 7 post vaccination. Animals that did not produce a take by day 7 post vaccination were re-vaccinated with either 5.7 log_10_ PFU of TNX-801 (low dose group) or 5.4 log_10_ PFU of rVACV.

### 2.6. Intratracheal Challenge with Monkeypox Virus

Sixty days after vaccination, all animals were administered a challenge dose of 5.0 log_10_ PFU/NHP (in 1.0 mL inoculum volume) of monkeypox virus (MPXV strain Zaire (V79-I-005; BEI Research Resources) via intratracheal inoculation. All animals were observed at least once daily (or more frequently if warranted) following the challenge with MPXV.

### 2.7. Body Weight, Lesion Counts, and Swabs

Body weight was measured during the challenge period (every 3 days). Body weights were determined starting at day 0 post challenge. Post-challenge body weight was determined every three days beginning on days 60 through 81, and days 85 and 88 (time points indicated in Figure 2). Post-challenge percent body weight change with respect to day 60 before vaccination was calculated and compared for all the animals. Lesion counts, oral swabs, blood collections, and photos were performed every three days between days 60 through 81, and on days 85 and 88.

### 2.8. Viral Load by Quantitative PCR (qPCR) Analysis

Whole blood and oral swabs were collected according to time points represented in Figure 2. Total DNA was extracted from each sample using QIAcube (Qiagen, Germantown, MD, USA) robot according to manufacturer protocol and viral genome copies were quantified via qPCR. The qPCR reaction was conducted using QuantStudio 6 Flex according to the manufacturer’s protocol in the format of 10 µL PCR reaction with at least two replicates. A pan-orthopoxvirus qPCR assay was carried out as previously described [25] using the following primers and probe: forward primer—5′-GAACATTTTTGGCAGAGAGAGCC-3′, reverse primer—5′-CAACTCTTAGCCGAAGCGTATGAG-3′, probe—6′FAM/CAGGCTACC/ZEN/AGTTCAA/3IABkFQ-3′. Briefly, the reaction was carried out under the following conditions: 50 °C for 2 min (1 cycle), then 95 °C for 1 min (1 cycle) and then 45 cycles of 15 s at 95 °C, and for 15 s at 63 °C. A melting curve analysis was performed at the end of the amplification. The dilutions were tested in duplicates and used as quantification standards to construct the standard curve. The amount of DNA quantified for each sample was expressed as the number of genome copies/reaction.

### 2.9. Enzyme-Linked Immunosorbent Assay (ELISA)

Serum samples were collected during the vaccination phase as per the time point shown in Figure 3A and were used to determine antibody titers to VACV. Briefly, serum was collected prior to vaccination (Study Day-3) and on Study Days 28 and 56. In order to measure vaccinia virus (VACV)-specific antibody responses during the vaccination and challenge phase, serum samples were analyzed using a VACV-specific ELISA in accordance with Southern Research Standard Operating Procedure. Briefly, 0.5 μg/mL of purified inactivated VACV (Western Reserve (WR)) particles was coated onto 96-well in phosphate buffered saline (PBS) overnight at 2–8 °C. An automated plate washer (BioTek ELx405, Winooski, VT, USA) was used to remove any unbound antigen and then the wells were blocked in 5% non-fat milk in 0.05% PBS Tween 20 (PBST) at 37 ± 1 °C. Serum samples were serially diluted in 5% non-fat milk in 0.05% PBST and added at 100 uL/well. Specific antibodies were detected using a goat anti-monkey horseradish peroxidase (HRP) conjugated IgG as secondary antibody. Plates were developed using 100 μL/well of 2,2′-Azinobis [3-ethylbenzothiazoline-6-sulfonic acid]-diammonium salt (ABTS) substrate for 15 to 20 min, stopped with 100 μL of 1% sodium dodecyl sulfate (SDS) in distilled water and read at 405 nm using a SpectraMax plate reader (Molecular Devices. Sunnyvale, CA, USA). Positive (monkey polyclonal antibody) and negative (normal serum) control samples were included on each plate. The plates were read using an optical density of 450 nm. The highest fold dilution at which the optical density of a sample is ≥0.2 is reported as the titer value.

### 2.10. MPXV Plaque Reduction Neutralization (PRNT) Assay

For PRNT, serum was collected prior to vaccination (Study Day 3) and on Study Days 28 and 56. Following the challenge, additional blood samples were collected for immunoassays on post-challenge Days 6, 12, and 28 (Figure 2). The ability of VACV ELISA-binding antibodies to neutralize MPXV was tested using a validated MPXV-specific PRNT assay in accordance with Southern Research standard operating procedures. Briefly, heat-inactivated serum samples were serially diluted in DMEM containing Glutamax and 2% FBS and added to an equal volume of a fixed dilution of MPXV (Zaire strain). The serum–virus mixture was then incubated overnight at 2–8 °C. One hundred microliters of each sample was added to a 24-well plate containing Vero cells in triplicate and incubated at 37 ± 1 °C. MPXV pre-incubated with a normal monkey and immune serum served as negative and positive controls, respectively. Neutralization endpoint titers were calculated by determining the reciprocal dilution of serum necessary to inhibit 50% of the plaque formation when compared to the control virus sample. The neutralization was calculated as the percentage of the number of plaque counts reduced in the testing serum per assay compared to the mean number of the plaque counts for the controls in the same assay.

### 2.11. Statistical Analysis

All statistical analyses were performed in GraphPad Prism version 9 for Windows, GraphPad Software, San Diego, CA, USA, www.graphpad.com” using built-in functions. Ordinary one-way ANOVA, followed by Tukey’s multiple comparison test, was used to compare three or more groups. Differences were considered statistically significant when *p* was <0.0001; *** *p* < 0.001; ** *p* < 0.01; * *p* < 0.05).

## 3. Results

### 3.1. TNX-801 Immunogenicity

To evaluate the immunogenicity of TNX-801, four groups of NHPs (N = 4/group) were vaccinated at Day 0 via scarification using a bifurcated needle with TNX-801 approximately 6.6 log_10_ PFU (high dose), 5.7 log_10_ PFU of TNX-801 (low dose), as a control, we chose to vaccinate monkeys with 5.0 of log_10_ PFU of rVACV, or diluent (mock) (Figure 2). Following vaccination, NHPs were monitored daily for skin lesions. Regardless of the vaccination status, none of the NHPs displayed any lesions for 60 days post vaccination. A single NHP was removed from the rVACV group due to unrelated health issues.

In susceptible individuals, vaccination with live replicating vaccinia virus-based vaccines produces a major cutaneous reaction at the inoculation site approximately 7 days post vaccination (referred to as a “take”) [13,26,27]. The development of a take was assessed 7 days post vaccination. All animals vaccinated with the high dose of TNX-801 and three of the four animals vaccinated with the low dose of TNX-801 exhibited a take at day 7. In the rVACV arm, only one of the four animals exhibited a take. Any animals that did not present a take were re-vaccinated on day 14 with the following doses: 5.7 log_10_ PFU of TNX-801 (low dose) and 5.4 log_10_ PFU of rVACV. A take was observed in all re-vaccinated animals except NHP #3 (rVACV Group 3).

Humoral responses were measured via testing the binding serum antibodies in rVACV ELISA and MPXV PRNT_50_ assays. Total IgG responses from serum at days 28 and 56 post vaccination demonstrated that all eight NHPs in both TNX-801 vaccine groups had titers 2- to 16-fold higher than the respective baseline values (Figure 3A). Animals vaccinated at the higher dose (6.6 log_10_ PFU) of TNX-801 exhibited an increase in anti-VACV antibody titer ranging from 2- to 16-fold by Day 28 and 4- to 8-fold by Day 56 (Figure 3A). The increase in antibody titer for the lower dose group (5.7 log_10_ PFU) of TNX-801 ranged from 4- to 8-fold by Day 28 and 4- to 16-fold by Day 56. In contrast, the rVACV group yielded titers ranging from 1- to 4-fold higher than baseline values at both Day 8 and Day 56.

Neutralizing antibody responses to MPXV displayed a similar pattern to ELISA responses. Seven of the eight NHPs in TNX-801 groups exhibited neutralizing antibody responses with increases in PRNT_50_ titers ranging from 8- to 50-fold from their respective baseline values (Figure 3B). In contrast, only two of the three NHPs in the rVACV group had neutralizing antibody responses with titers of 6- to 34-fold higher. This lower neutralizing antibody response in the rVACV group may be due in part to the lower initial dose of virus administered to each animal compared to the TNX-801 (low dose) group.

### 3.2. TNX-801 Vaccination Efficacy in NHPs Following Lethal Challenge with MPXV (Zaire)

NHPs were challenged at 60 days post vaccination with 5.0 log_10_ PFU of MPXV (Zaire) via the IT route. Following the challenge, NHPs were monitored for virus shedding (throat swabs), viremia, weight loss, skin lesions, and severe clinical disease.

Virus shedding at the site of the challenge was investigated via throat swabs at three-day intervals until 21 days post infection (dpi) and four-day intervals afterwards until 28 dpi (Figure 4A). Six of the eight NHPs in both groups receiving the TNX-801 vaccine had undetectable levels of viral genome throughout the 28-day sampling period. Only two NHPs, NHP #4 in TNX-801 (high dose) and NHP #3 in TNX-801 (low dose), had detectable levels of viral DNA at single time points. NHP #4 and NHP #3 had 4.4 and 3.1 log_10_ copies per swab, respectively (Figure 4A). In contrast, rVACV and mock vaccinated groups had high levels of viral DNA that were detected throughout 28 dpi. The NHP #1 in the rVACV group had viral DNA present at all time point after 6 dpi with peak value of 7.3 log10 copies per swab. NHPs #3 and 4 had detectable viral DNA at three time points with peak values of 5.0 and 5.1 log10 copies per swab. The mock vaccinated group had the highest viral DNA levels following the challenge. Viral DNA was detected in three of the four NHPs as early as 6 dpi and remained high at all following time points with peak values ranging from 5.8 to 8.5 log10 copies per swab.

Viremia in whole blood was detected via qPCR with an identical schedule to throat swabs. Three of the four NHPs in the TNX-801 (high dose) group had sporadic viremia at one to two timepoints during the 28-day sampling period (Figure 4B). In this group, NHP #2 had detectable levels of viral DNA at 6 and 9 dpi with a peak value of 6.4 log_10_ copies/mL. Viremia was detected at a single timepoint at 18 dpi for NHP #3 with a value of 4.9 log_10_ copies/mL. All NHPs vaccinated with TNX-801 (low dose) had no detectable level of viremia at any timepoints. Two of the three NHPs in the rVACV group had detectable levels of viremia with peak values ranging from 6.0 to 6.9 log_10_ copies/mL. In contrast to the vaccinated groups, the mock-treated group had the highest magnitude and duration of viremia. Viral DNA could be detected as early as 3 dpi in NHP #3, followed by a rapid increase in viral load in all NHPs. Peak viral loads in NHPs ranged from 7.3 to 9.7 log_10_ copies/mL.

Weight loss in NHPs following the challenge was monitored with an identical schedule to virus shedding and viremia. Seven of the eight NHPs in the TNX-801 groups had either no or minimal weight loss. NHP #2 in TNX-801 (high dose) group rapidly lost weight starting at 3 dpi with a peak decline of −16% at 9 dpi (Figure 4C). The NHP was not able to regain weight comparable to baseline throughout the remainder of the study (Figure 4C). A similar pattern was observed in the weight of animals receiving the rVACV vaccine. Two of the three NHPs exhibited minimal or no weight loss (Figure 4C). NHP #1 lost weight starting at 3 dpi with peak loss of −11% at 6 dpi and was not able to regain weight comparable to baseline. In contrast, all four NHPs rapidly lost weight starting at 3 and 6 dpi with peak weight loss ranging from 3.4 to 13%.

All NHPs in the mock group developed progressive cutaneous lesion development following the MPXV challenge (Figure 4D). NHPs displayed lesions starting at 9 dpi with a rapid rise in lesions throughout the face and/or torso (Figure 4D and Figure 5A,B). No pock lesions were detected in any of the animals vaccinated with TNX-801 prior to MPXV challenge (Figure 4D and Figure 5B). Two of the three animals in the rVACV group exhibited lesions at 9 dpi. NHP #4 had 19 lesions at 9 dpi followed by a rapid clearance. In contrast, NHP #1 had a sustained lesion outbreak that did not fully resolve by 28 dpi. Two of the four animals in the control group exhibited severe disease and met the euthanasia criteria at 9 and 12 dpi (Figure 5C). All NHPs in the TNX-801 vaccinated groups survived the lethal challenge with few or no signs or symptoms of poxvirus infection (Figure 5C).

## 4. Discussion

Molecular studies suggest that modern VACV strains share a common ancestry with HPXV. Both genomic and phylogenetic analyses revealed that the core genome of the Mulford 1902 smallpox vaccine had 99.7% similarity to HPXV [16]. A U.S. vaccine from circa 1860 also had >99.9% colinear identity [18]. Furthermore, historical reports indicate that Jenner and others used material from infected horses (termed “grease”) extensively in the prevention of smallpox from Jenner’s time until the end of the 19th century [11,28,29]. Here, we report the development TNX-801, a rcHPXV-based vaccine.

The TNX-801 was readily amplified and purified in cell culture. The vaccination in NHPs was well tolerated as indicated by the lack of any overt clinical disease. The single dose of TNX-801 was able to generate robust humoral responses as measured by the total IgG and neutralizing response. Following the challenge, the vaccinated NHPs showed little or no virus shedding, viremia, and weight loss. No TNX-801 vaccinated NHPs showed any sign of severe clinical disease or any lesions throughout the 28-day study and survived the challenge. These data show that the immunogenicity generated by TNX-801 was able to provide protection against the most lethal challenge strain of MPXV (Zaire) and is the first study to demonstrate the efficacy of TNX-801 vaccination against MPXV challenge in a non-human primate model.

One limitation of our study is the dose administered to vaccinate the cynomolgus macaques for rVACV was lower than both doses of TNX-801, making it difficult to compare humoral response vaccine efficacy in animals vaccinated with TNX-801 to rVACV. This proof-of-concept study, however, clearly demonstrates that TNX-801 can be used as a vaccine to protect NHPs from a lethal MPXV challenge, and warrants further investigation as a potentially safer and more effective vaccine than what is currently available to the public. Additional studies are required to better define the innate and adaptive (T cell and humoral) immune responses following vaccination with TNX-801.

Smallpox eradication in 1980 via vaccination (Dryvax) was a seminal accomplishment in the field of vaccinology. However, due to the potential utilization of smallpox in the context of bioterrorism, two safer vaccines have been developed and approved for human use, ACAM2000 and Jynneos. ACAM2000 is a live attenuated vaccine derived from Dryvax via plaque pick; however, the vaccine is potentially associated with severe adverse events such as myocarditis. The Jynneos vaccine is a replication-deficient MVA virus that requires two subcutaneous vaccinations 30 days apart to produce comparable humoral immunity to ACAM2000. Both vaccines offer potential drawbacks. One important drawback is vaccination in the context of a ring vaccination setting. Ring vaccination was the first-line strategy to effectively contain and limit smallpox outbreaks, eventually leading to the eradication of smallpox. In this scenario, a single dose vaccine with a safer profile than Dryvax or ACAM2000 would be considerably beneficial to limit and control smallpox outbreaks via bioterrorism and/or natural monkeypox outbreak.

Another potential drawback of MVA relates to the question about the functional equivalence of neutralizing antibody responses generated by MVA-BN as compared to replicating vaccinia vaccines in NHPs [30] and in humans [9]. Although the replication-deficient MVA vector is generally safer compared to replication-competent Dryvax or ACAM2000, it requires two doses of high-titered vaccine to induce a sufficient immune response in humans [9]. While MVA is believed to be safer than ACAM2000 in immunodeficient individuals, the relatively small number of MVA-vaccinated individuals is not enough to determine if there is a safety advantage in immunocompetent individuals, for example, with respect to cardiomyositis. Moreover, MVA was not used widely in the eradication effort, so it is not clear whether it would provide equivalent protection in generating herd immunity, for example, by blocking forward transmission or providing durable (years or decades) levels of protection.

The current monkeypox outbreak throughout Africa, Europe, and North America has demonstrated again that pathogenic poxviruses can spread rapidly around the globe and cause severe clinical disease. Collectively, these data strongly demonstrate the need to develop safer smallpox and monkeypox vaccines to combat future outbreaks.

In summary, a single dose vaccination with TNX-801, a replication competent HPXV vaccine, was effective at protecting NHPs from infection with MPXV. TNX-801 vaccination prevented MPXV lesion formation as well as signs of clinical disease, which could have significant implications for protecting both individuals and populations in the event of the natural or terror-inspired re-emergence of smallpox. The current positive, but proof-of-concept preliminary data on using TNX-801 as a vaccine against MPXV are encouraging and supportive of further development of this platform.

## 5. Patents

Tonix Pharmaceuticals holds relevant patents pertaining to the use of synthetic virology technologies associated with this work.

## Figures and Tables

**Figure 1 viruses-15-00356-f001:**
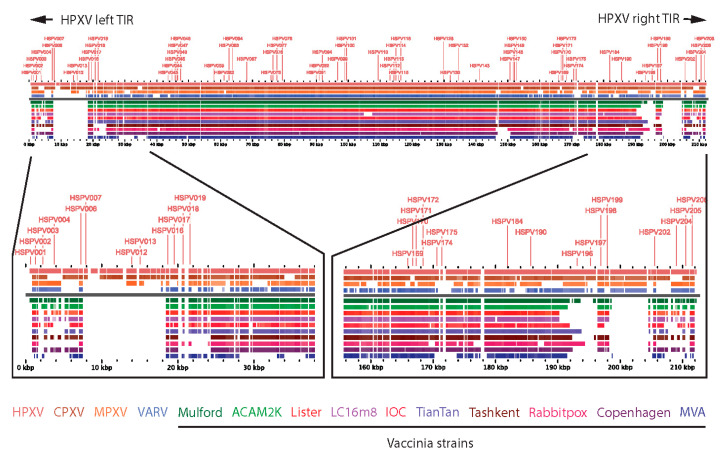
Alignment of orthopoxvirus genomes. Orthopoxvirus genomes were aligned using the program GView (https://server.gview.ca; accessed on 24 October 2022). A BLAST alignment was performed to display similarities within the gene sequences of the HPXV reference genome (NCBI Accession KY349117) and the following orthopoxvirus genomes (CPXV—KC813505; MPXV—ON676705; VARV Bangladeshi 1975—DQ437586; VACV Mulford 1902—MF477237; VACV ACAM2K—AY313847; VACV Lister—DQ121394; VACV LC16m8—AY678275; VACV IOC-B141—KT184690; VACV TianTan—JX489136; VACV Tashkent—KM044310; RPXV Utrecht—AY484669; VACV Copenhagen—M35027.1; MVA—AY603355). The actual nucleotide sequence of each gene within the genome was compared to the coding sequence (CDS) of each gene within the HPXV genome. The white gaps in the HPXV reference sequence represent non-coding sequences within the genome. The percent identity (PID) was set to 85%, meaning that only hits with PID values over the selected cutoffs were displayed.

**Figure 2 viruses-15-00356-f002:**
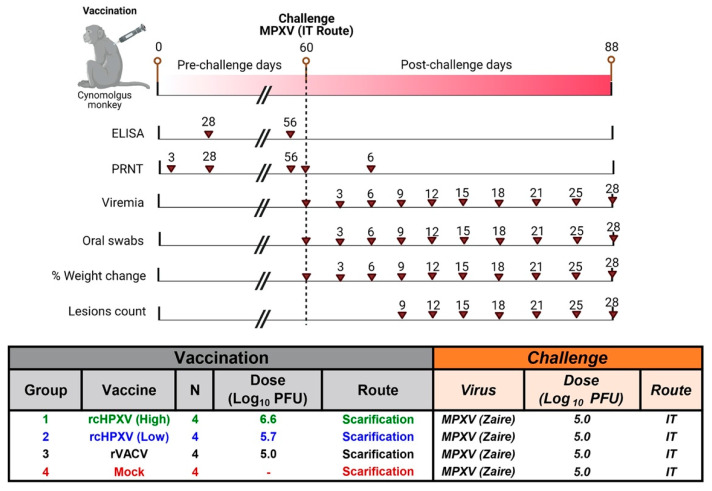
Schematic design of the NHP study. Four cynomolgus macaques per group were vaccinated with rcHPXV, rVACV, or diluent. Following vaccination, immunogenicity was evaluated at intervals indicated by upside down triangle. AT 60 days post-vaccination, NHPs were challenged with MPXV (Zaire) via IT route. Following the challenge, viremia, virus shedding, weight loss, lesion count, and survival was determined.

**Figure 3 viruses-15-00356-f003:**
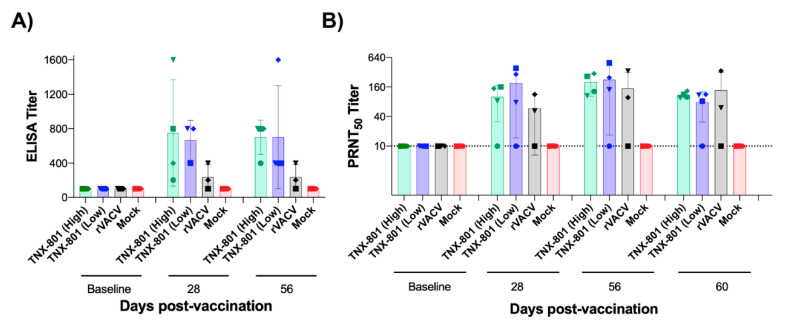
Immunogenicity in NHPs following vaccination. Total IgG and neutralizing antibodies were determined by ELISA (**A**) and PRNT_50_ assay (**B**). Limit of detection (LOD) is shown via dashed line. ELISA LOD = 100. PRNT_50_ LOD = 10.

**Figure 4 viruses-15-00356-f004:**
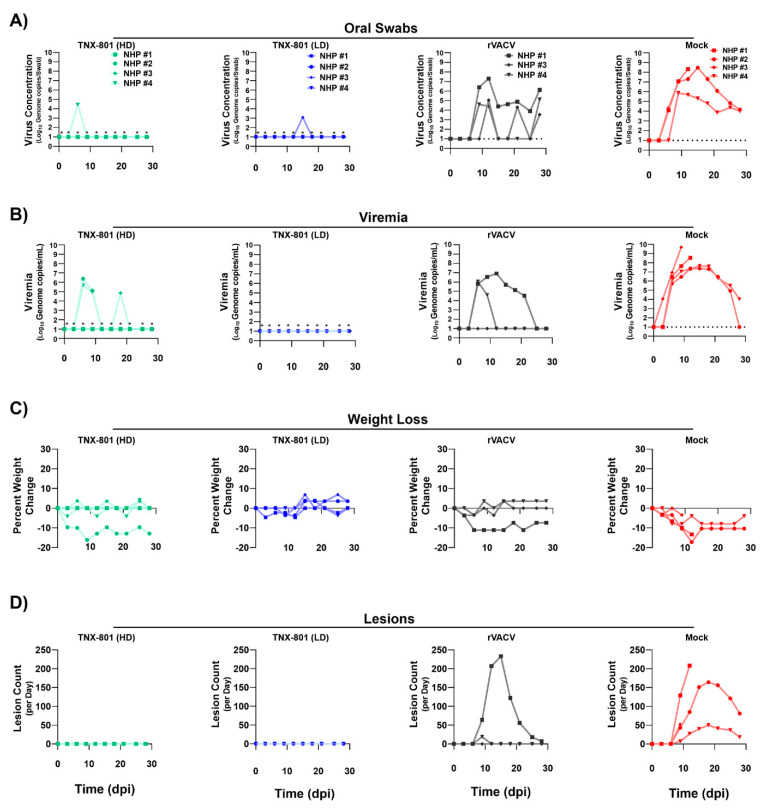
Determination of clinical MPXV disease characteristics in vaccinated groups of cynomolgus macaques following lethal MPXV (Zaire) challenge. Animals were monitored daily for the presence of clinical disease markers, including (**A**) virus shedding in oral swabs and (**B**) viremia in blood by qPCR, (**C**) weight loss, and (**D**) lesion counts. Groups of cynomologus macaques are as follows: high dose (HD) TNX-801, low dose (LD) TNX-801), rVACV, and Mock. Limit of detection (LOD) is shown via a dashed line. LOD for MPXV qPCR = 1 log_10_ Genome copies/mL. *p*-values < 0.01 are indicated by *.

**Figure 5 viruses-15-00356-f005:**
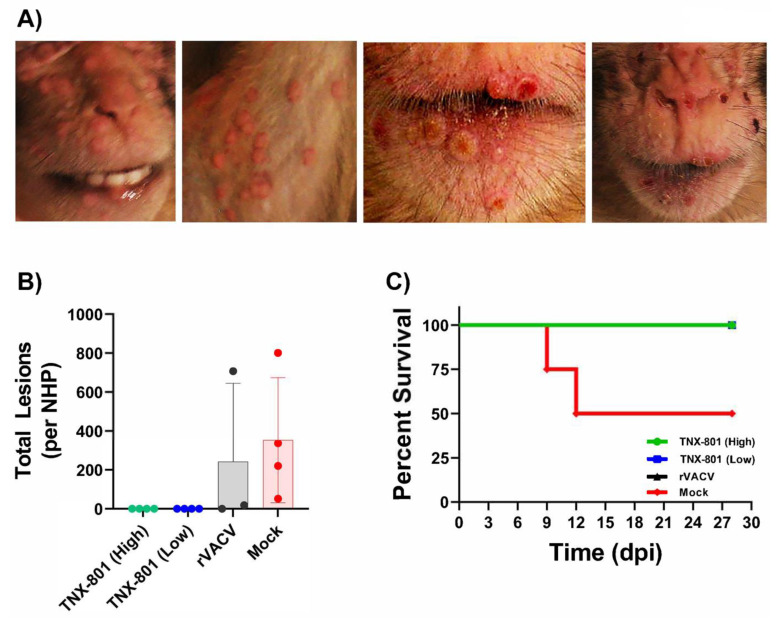
Clinical disease and survival in NHPs following lethal MPXV (Zaire) challenge. Photograph of lesions (**A**) and total lesion counts (**B**) caused by MPXV are shown. Percent survival of NHP is shown (**C**).

## Data Availability

Not applicable.

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
