# Peer review of "Single Dose of Recombinant Chimeric Horsepox Virus (TNX-801) Vaccination Protects Macaques from Lethal Monkeypox Challenge"

_viruses, 2023, doi:10.3390/v15020356_

Round 1

Reviewer 1 Report

The manuscript entitled "Single dose of Recombinant Chimeric Horsepox Virus (TNX-801) Vaccination Protects Ma-caques from Lethal Monkeypox Challenge" reports use of HPXV as vaccine for MPXV. It is important to develop a vaccine for MPXV. The manuscript can be improved in following aspects:

1. Limitations of the study should be included.

2. A proper comparison with existing vaccines is lacking. Authors may compare (literature comparison) the immunogenicity of the proposed vaccine with previous reports.

Minor comments:

1. First sentence should be stated about MPXV (not for smallpox).

2. "The vaccine was well tolerated and protected from lesions or severe disease". vaccine? Please correct it.

3. Provide more details about qPCR especially primers.

4. Fig.3 A. Standard deviation is very high. Did authors perform statistical analysis?

5. Write expanded form of all abbreviations at first use. For example, dpi etc.

6. Figure 4. Explain each subplot in figure captions. It is hard to understand.

Reviewer 2 Report

Please address the following questions and suggested revisions:

1. The ACAM2000 is available via the CDC. The experimental design would have benefited from inclusion of the ACAM2000 (or Jynneos) vaccine. Please address why ACAM2000 was not included.

2. The majority of the animals vaccinated with the rVACV and a single animal vaccinated with the experimental horsepox vaccine (TNX-801) did not produce a vaccine "take" and the vaccination was repeated on Study Day 14. Please note this in the experimental design figure (Fig. 2) and discuss whether this affected the outcome for these particular animals. For example, was viral load and pox counts markedly different in the animals that received the second vaccination vs the other animals in these groups. Also, one animal in the rVACV group did not have a "take" even after the second vaccination. Did this animal have a suboptimal outcome after challenge?

3. Fig. 2 shows viremia and VL sampling until study Day 12 but Fig. 4 shows VL data for later time points. Please correct Fig. 2 accordingly. 

4. Please list the source of the MPXV challenge strain (Zaire) and include the volume used for the intratracheal inoculation. Please address why the IT route was chosen (vs IV).

5. It is noted that the animals were observed "at least once daily". Once daily seems infrequent for a study which includes severe illness in the animals. Were these observations conducted cage-side by a trained animal technician. Did any of the animals require supportive treatment (anti-inflammatory, modified diet, etc.)?

6. Fig. 3 A) lists "baseline" vs 3 B) which lists Day 3 for serology analyses. I believe the Day 3 should be revised to Day -3 and this time point was also used for A). Please revise to either "Baseline" for both or Day -3 for both graphs.

7. For plaque assay, please list the acceptance criteria for the plaque counts in the control wells (virus only) and the titer of the pos. control.

8. For the qPCR analysis, the primer/probe sequences were not included. Please either list these or include an appropriate reference.

9. Fig. 4 does not include group information. Please add these to the legend or above the graphs. Fig. 4A) and D) seem to be missing an animal for the last group listed (mock control).

10. It is mentioned on page 10 that one of the animals in the high dose group lost a significant amount of weight. Did this correlate with viral loads and clinical signs for this animal? Did this animal have reduced levels of antibodies? Please discuss.

11. What was used as euthanasia criteria? Was supportive treatment provided for any of the animals?

Reviewer 3 Report

Nice manuscript describing a "synthetic" vaccine (horsepox virus) and its efficacy to prevent clade I monkeypox virus infection (intratracheal challenge)  as a one dose vaccine. Control vaccine arm, vaccinia virus, was used at a dose less than what is licensed, and at a lesser dose than the recombinant horsepox virus vaccine. 

Benefit of horsepox virus vaccine to prevent monkeypox infection is demonstrated in the animal challenge model. 

Authors speculate on the enhanced safety of the horsepox virus vaccine; no data is provided to support that assertion. It is noted that horsepox virus vaccine does produce smaller plaques than the vaccinia virus vaccine. 

Reviewer 4 Report

This manuscript presents results of a proof-of-concept study of a proposed monkeypox vaccine based on a recombinant chimeric horsepox virus in an animal model. My comments are as follows:

A major issue is that the results present comparison of humoral response and vaccine efficacy of rVACV and TNX-801. However, the administered dose of rVACV is lower than both doses of TNX-801 (high and low). This does not seem like a fair comparison. How do the authors justify this comparison and account for the effect of lower dosage of rVACV? In addition, how do we know if an equal/higher dose of rVACV as that of TNX-801 will not have the same humoral response?

 The discussion section lines 378-380 raise concerns about the T cell response of MVA as compared to replicating vaccinia vaccines. Please provide citations for this.

Discussion Section lines 391-393, no references are provided of the 'growing body of evidence'. This complete point about T cell responses also seems incomplete as

the authors do not provide any evidence of t cell responses generated in response to TNX-801. How do we know the T cell profile of TNX-801 is better/similar/lower than MVA? Authors should properly back this statement with sufficient references.

Line 270. Is this a typo? If not, then sVACV is undefined.

Round 2

Reviewer 1 Report

Revised version is much improved and can be accepted for publication.

Reviewer 4 Report

The authors have address the comments I raised in the previous round of review. I recomend this manuscript for publication.